# SiamCAN: Simple yet Effective Method to enhance Siamese Short-Term Tracking

## Abstract

Most traditional Siamese trackers are used to regard the location of the max response map as the center of target. However, it is difficult for these traditional methods to calculate response value accurately when face the similar object, deformation, background clutters and other challenges. So how to get the reliable response map is the key to improve tracking performance. Accordingly, a simple yet effective short-term tracking framework (called SiamCAN),by which bridging the information flow between search branch and template branch, is proposed to solve the above problem in this paper. Moreover, in order to get more accurate target estimation, an anchor-free mechanism and specialized training strategy are applied to narrow the gap between the predicted bounding box and groundtruth. The proposed method achieves state-of-the-art performance on four visual tracking benchmarks including UAV123, OTB100, VOT2018 and VOT2019, outperforming the strong baseline, SiamBAN, by $0.327 \rightarrow 0.331$ on VOT2019 and $0.631 \rightarrow 0.638$ success score, $0.833 \rightarrow 0.850$ precision score on UAV123.

## 1 Introduction

Visual object tracking is the fundamental task of computer vision, aiming at tracking unknown object of which the information is given by the first frame. Although great progress has been achieved in recent years, a robust tracker is still in desperate demand due to tricky challenge such as scale variation, appearance deformation and similar object with complex background which can deteriorate tracking performance (Wu et al. (2013); Zhang et al. (2014)).

Recently, Siamese Network based trackers have taken a vital place in SOT field due to its accuracy and speed. Since (Tao et al. (2016)) and (Bertinetto et al. (2016)) introduced Siamese networks in visual tracking, Siamese structure has been adopted as baseline for researchers to design efficient trackers (Li et al. (2018); Zhu et al. (2018a); Zhang & Peng (2019); Li et al. (2019); Xu et al. (2020); Chen et al. (2020)). After siamRPN (Li et al. (2018)) being proposed to gain more accurate anchor boxes, region proposal network has become an essential part of tracker. However, the anchor scales are manual-set which go against the fact that the tracking target is unknown. Besides, the performance of the Siamese based trackers depends greatly on offline training by using massive frame pairs. Therefore, it highly increases the risk of tracking drift when facing significant deformation, similar object distractors, or complex background, due to the undiscriminating feature learned from the target when the category of the target is excluded from the training dataset.

In these years, the attention mechanism has become the spotlight in computer vision which inspires the relative works not only in detection task but also in visual tracking (He et al. (2018); Abdelpakey et al. (2018); Wang et al. (2018); Zhu et al. (2018b)). The attention mechanism includes channel attention and spatial attention, the former tends to generate a set of channel-weights for modeling interdependencies between channels while the latter focuses on finding the informative part by utilizing the inter-spatial relationship of features. Considering these benefits, Siamese based trackers try to introduce attention module to distinguish target from complex background. Nevertheless, the performance of these trackers is not satisfactory for exploiting the expressive power of the attention mechanism inappropriately.

Based on the limitations discussed above, we design a simple Cross-attention Guided Siamese network (SiamCAN) based tracker with anchor-free strategy which performs better than the state-of-the-art trackers when facing the similar object challenge. SiamCAN takes template channel attention

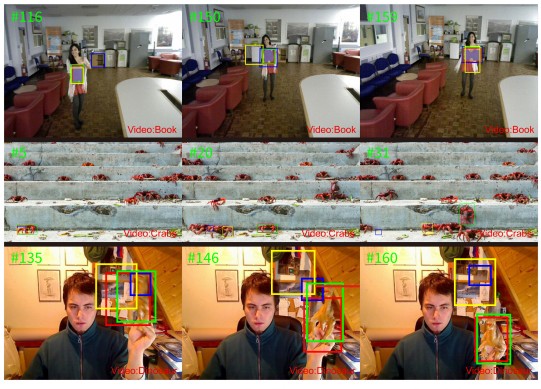

Figure 1: Tracking results on the challenges of three aspects: scale variation (book), similar distractors (crabs) and fast motion (dinosaur). Compared with other two state-of-the-art trackers, our tracker (SiamCAN) performs better.

to guide the feature extraction of search image by which can strengthen the ability of tracker to overcome distractors and complex backgrounds, performing better than most of Siamese-based trackers, as shown in Figure 1. The main contributions of this work are:

- We formulate a cross-attention guided Siamese framework (SiamCAN) including cross-channel attention and self-spatial attention. The cross-channel attention builds an interactive bridge between the target template and search frame to share the identical channel weights. The self-spatial attention focuses on the discriminative part of the correlated feature map, which is complementary to the cross-channel attention.

- The proposed tracker is adaptive box regression, without numerous hyper-parameters setting. In order to get more accurate bounding box, we adopt the proper strategy to utilize the merits of anchor-free at the stage of training.

- SiamCAN achieves state-of-the-art results on four large tracking benchmarks, including OTB100 (Wu et al. (2013)), UAV123 (Mueller et al. (2016)), VOT2018 (Kristan et al. (2018)) and VOT2019 (Kristan et al. (2019)). The speed of tracker can also achieve 35 FPS.

## 2 RELATED WORK

In this section, we briefly review the recent Siamese based trackers, the anchor-free approaches and attention mechanism in both tracking and detection filed.

### 2.1 SIAMESE NETWORK BASED TRACKER

The pioneering works, SINT (Tao et al. (2016)) and SiamFC (Bertinetto et al. (2016)), first introduce the siamese network in tracking filed. Due to its fast speed with light structure, Siamese network draws great attention from the visual tracking community. SiamFC tries to use siamese network to learn the feature of both target template and search frame, and compare the similarity of them to find the most confident candidates. Although tracks fast, it cannot handle the scale variation problem by applying several scales of feature map. Inspired by Faster-RCNN (Ren et al. (2015)) from object detection, SiamRPN (Li et al. (2018)) draws on the region proposal network to get more various scale ratio bounding boxes. Since then, the RPN module has become an essential part of the tracker (Zhu et al. (2018a); Zhang & Peng (2019); Li et al. (2019); Dong & Shen (2018); Fan & Ling (2019)). However, the complexity of anchor design makes the performance of trackers depend greatly on the effect of anchor training.

## 2.2 ANCHOR-FREE APPROACHES

In recent time, Anchor-free approaches have developed fast. The achor-free work can be divided into two categories. The first one (Kong et al. (2019); Law & Deng (2018)) aims to estimate the keypoints of the objects, while, the other (Redmon et al. (2016); Tian et al. (2019))tends to predict the bounding box for each pixel which can avoid presetting the scale ratio of anchors. Not only is anchor-free approach popular in detection field, but it is suitable for target estimation in tracking field due to its high efficiency. SiamFC++ takes example by FCOS (Tian et al. (2019)) to design regression subnetwork and add centerness branch to eliminate the low quality samples. SiamCAR (Guo et al. (2020)) changes the basic network structure additionally, merging the multi-layers features before correlation. Different from SiamCAR, SiamBAN (Chen et al. (2020)) puts emphasis on the label assignment which improves the tracking performance. Our method differs from the above trackers in details (Section4.3).

## 2.3 ATTENTION MECHANISM

Attention mechanism has been the focus of the detection filed, on account to its powerful ability of enhancing deep CNNs. SE-Net (Hu et al. (2018)) firstly puts forward the mechanism to generate channel weights in return to direct the learning of channel attention. After that, CBAM (Woo et al. (2018)) utilizes both max-pooling and average-pooling to generate the merged attention, includes channel and spatial attention. Recently, ECA-Net (Wang et al. (2020b)) finds that avoiding dimensionality is of great importance for channel attention learning, and propose a cross-channel interaction strategy which performs better than SE-Net. In the tracking field, the recent trackers began to equip with the attention mechanism to get better performance. SA_Siam (He et al. (2018)) simply combines the SE-Net and SiamFC to get both discriminative and general features which boost the tracking performance. RASNet (Wang et al. (2018)) designs residual attention, general attention and channel attention to learn target feature better. SATIN (Gao et al. (2020)) uses hourglass network as backbone and designs a cross-attention module for exemplar branch to combine the channel and spatial attention from shallow and deep layers. However, these trackers only calculate alongside each branch and neglect the information flow between them, as a result, the ability of attention mechanism can not be fully utilized.

# 3 OUR APPROACH

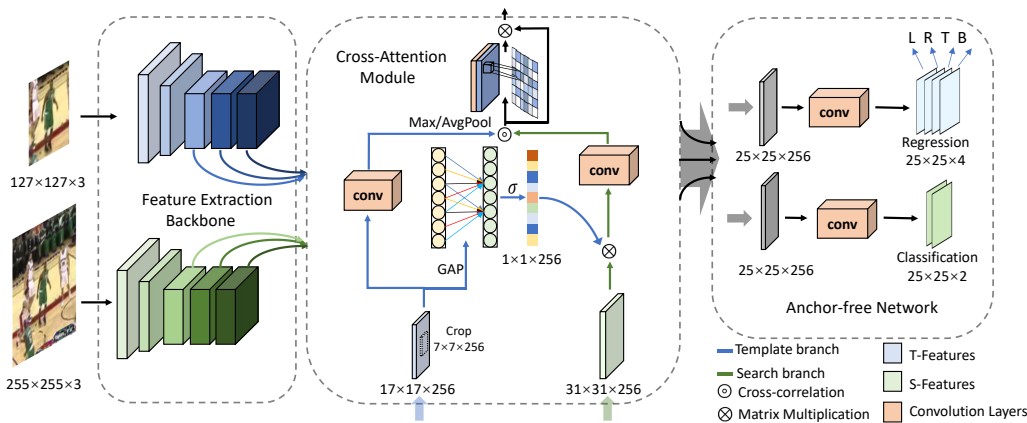

Figure 2: Illustration of the proposed tracking framework, consisting of feature extraction, cross-attention module and anchor-free network. In the feature extraction part, the features of the third, fourth and fifth blocks are sent to the cross-attention module. The cross-attention module uses the channel attention from the template branch to combine with the feature from the search branch. The right side shows the classification and anchor-free regression network, which is taken to localize the target and predict the size of bounding box.

As shown in Figure 2, the proposed framework mainly consists of three components, the feature extracted Siamese network, the cross-attention module and anchor-free bounding box regression subnetwork with foreground classification subnetwork.

## 3.1 Feature extracted Siamese Network

Like the most Siamese based tracker, we adopt the fully convolution network without padding, which guarantees the accurate location calculation. Feature extracted network composes of two parts, template branch and search branch. Both of them share the same layer parameters of backbone, by this mean, CNNs can learn the relative feature for them to calculate similarity in the subsequent operation. The template branch intends to encode the exemplar feature in the first frame while the other branch aims to encode the candidates feature which may involve target in the follow-up frames. Set the input in template branch as $I_t$, the subsequent frames in search branch as $I_s$. We feed the $I_t$ and $I_s$ into backbone, and get the feature $\phi_l(I_t)$ and $\phi_l(I_s)$ from different l-th backbone layers. Next, the given features are sent to the corresponding branch after having a convolution with a neck layer to reduce feature channel size to 256 and get the template feature $\psi_t(I_t)$ with the search feature $\psi_s(I_s)$. At last, crop the 7×7 patch from the center of template feature.

## 3.2 Cross-attention Network

Attention mechanism is created for the purpose that enforce CNNs to focus on the special parts which is of great importance, i.e., channels information and spatial information. The channel attention is designed to explore the interdependencies between channels while the spatial attention tends to make CNNs pay more attention to the most critical areas of the feature. Different from (He et al. (2018); Wang et al. (2018)), the channel attention is used between two branches not being applied as self-attention. Moreover, SATIN (Gao et al. (2020)) designs a module also called cross-attention, but the 'cross' means the combination of different layers which is different from our method. In this paper, the target branch feature $\psi_t(I_t)$ is sent to global average pooling to get aggregated feature $Y_t$, i.e.,

$$Y_t = \frac{1}{WH}\sum_{i=0}^{W,H}\psi_t(I_t) \tag{1}$$

Given the aggregated feature, the channel weight is obtained by performing a 1D convolution of size k, i.e.,

$$V_i = \sigma(\sum_{j=1}^{k}\omega^j y_i^j), y_i^j \in \Omega_i^k \tag{2}$$

Where $\sigma$ is a Sigmoid function, $\omega$ indicates the parameters of 1D convolution and $\Omega_i^k$ indicates the set of $k$ adjacent channels of $y_i$. To let search branch learns the information from target template, we multiply the channel weights with the search feature, i.e.,

$$\widetilde{\psi_s}(I_s) = \psi_s(I_s) * V \tag{3}$$

## 3.3 Classification and anchor-free regression subnetwork

As shown in Figure 2, the correlation feature map is calculated by the depth-wise correlation operation between $\widetilde{\psi_s}(I_s)$ and $\psi_t(I_t)$, i.e.,

$$F_{w \times h \times c}^{cls} = \widetilde{\psi_s}(I_s) \star \psi_t(I_t) \tag{4}$$

$$F_{w \times h \times c}^{reg} = \widetilde{\psi_s}(I_s) \star \psi_t(I_t) \tag{5}$$

where $\star$ denotes depth-wise convolution operation. Then, we apply self-spatial attention to the feature map in order to focus on discriminative part automatically, i.e.,

$$\widetilde{F}_{w \times h \times c}^{cls} = \sigma(f([AvgP(F_{w \times h \times c}^{cls}); MaxP(F_{w \times h \times c}^{cls})])) \tag{6}$$

$$\widetilde{F}_{w \times h \times c}^{reg} = \sigma(f([AvgP(F_{w \times h \times c}^{reg}); MaxP(F_{w \times h \times c}^{reg})])) \tag{7}$$

After that, we use two convolution layers with kernel size 1×1 to reduce the number of channel from 256 to 2 and 4 respectively for each branch and concatenate the feature maps from different layers of backbone by the trainable weights $\alpha$, i.e.,

$$P_{w \times h \times 2}^{cls} = \sum_{l=1}^{N}\alpha_l * \widetilde{F}_{l:w \times h \times 2}^{cls} \tag{8}$$

$$P_{w \times h \times 4}^{reg} = \sum_{l=1}^{N} \alpha_l * \widetilde{F}_{l:w \times h \times 4}^{reg} \tag{9}$$

where $N$ denotes the total number of the backbone layers we use. The classification feature map has two channels, the one represents the foreground and the points $(i, j)$ on $P_{w \times h \times 2}^{cls}(0, i, j)$ refer to the probability scores of target, the other represents the background and the points $(i, j)$ on $P_{w \times h \times 2}^{cls}(1, i, j)$ refer to the probability scores of background. The regression feature map has four channels, each of them represents the four direction distances from the points location in search branch input to the four sides of the bounding box respectively, that is to say, each point $(i, j)$ in $P_{w \times h \times 2}^{reg}(:, i, j)$ is a vector which can denoted as $(l, r, t, b)$.

**Classification label and regression label.** For anchor based methods, the positive sample and the negative one are classified by the value of Intersection over Union between anchor and groundtruth. In this paper, We use ellipse and circle figure region to design label for points $(i, j)$ in feature map, which is inspired by (Chen et al. (2020)). The ellipse $E_1$ center and axes length are set by groundtruth center $(g_{x_c}, g_{y_c})$ of groundtruth size $(\frac{g_w}{2}, \frac{g_h}{2})$, We also get the circle $C_2$ with $\frac{0.5g_w * 0.5g_h}{((\frac{g_w}{2})^2 + (\frac{g_h}{2})^2)^{\frac{1}{2}}}$ as radius, i.e.,

$$\frac{(B(p_i) - g_{x_c})^2}{(\frac{g_w}{2})^2} + \frac{(B(p_j) - g_{y_c})^2}{(\frac{g_h}{2})^2} = 1 \tag{10}$$

$$B(p_i)^2 + B(p_j)^2 = r^2 \tag{11}$$

where $B$ denotes the calculation for the location of points $(i, j)$ in feature map $P_{w \times h \times 2}^{cls}$ back to the search frame. If the point $B(p_i, p_j)$ falls within the $C_2$ region, it will be given a positive label, and if it falls outside the $E_1$ area, it will be given a negative label, i.e.,

$$label = \begin{cases} 1, & if C_2(p(i,j)) < r^2 \\ -1, & if E_1(p(i,j)) > 1 \\ 0, & otherwise \end{cases} \tag{12}$$

For regression branch, the regression targets can be defined by:

$$d_{(i,j)}^l = p_i - g_{x_0}, d_{(i,j)}^t = p_j - g_{y_0} \tag{13}$$

$$d_{(i,j)}^r = g_{x_1} - p_i, d_{(i,j)}^b = g_{y_1} - p_j \tag{14}$$

where $(g_{x_0}, g_{y_0})$, $(g_{x_1}, g_{y_1})$ denote the left-top and right-bottom coordinates location of the groundtruth.

**Loss function.** We employ cross entropy loss to train the classification network. To predict more accurate bounding box, we adopt the DIoU loss (Zheng et al. (2020)) to train the regression network, i.e.,

$$L_{reg} = 1 - IoU + \frac{\rho^2(p, p^{gt})}{c} \tag{15}$$

where $\rho(.)$ is the Euclidean distance, $p$ and $p_{gt}$ denote the central points of predicted box and groundtruth and $c$ is the diagonal length of the smallest enclosing box covering the two boxes. For regression branch training, DIoU loss can optimize the bounding faster than GIoU loss (Rezatofighi et al. (2019)). The overall loss function is:

$$L = \lambda_1 L_{cls} + \lambda_2 L_{reg} \tag{16}$$

where constants $\lambda_1$ and $\lambda_2$ weight the classification loss and regression loss. During model training, we simply set $\lambda_1 = 1$, $\lambda_2 = 1$ without hyper-parameters searching.

## 3.4 TRAINING AND INFERENCE

**Training.** We train our model by using image pairs, a $127 \times 127$ pixels template patch and a $255 \times 255$ pixels search patch. The training datasets include ImageNet VID (Russakovsky et al. (2015)), COCO (Lin et al. (2014)), YouTube-Bounding Boxes (Real et al. (2017)), ImageNet Det (Real et al. (2017)) and GOT10k (Huang et al. (2019)). Due to the numbers of negative samples are more than the positive samples, we set at most 16 positive samples and 48 negative samples from the search image.

Besides, in order to get more accurate regression information, we adopt the DIoU loss to optimize the regression branch.

**Inference.** We feed the cropped first frame and the subsequent frames to the feature abstract network as template image and search image. Next, the features are sent to the cross-attention module and pass the classification branch and regression branch. After that, we get the classification map. The location of the highest score represents the most probable center of the tracking target. Then, we use the scale change penalty and the cosine window as that introduced in (Li et al. (2018)) to guarantee the smooth movements of target. According to the location $p$ of the final score, we can get the predicted boxes $B$, i.e.,

$$b_{x1} = p_i - d_l^{reg}, b_{y1} = p_j - d_t^{reg} \tag{17}$$

$$b_{x2} = p_i + d_r^{reg}, b_{y2} = p_j + d_b^{reg} \tag{18}$$

where $d_{l,r,t,b}^{reg}$ denotes the predicted values of the regression targets on the regression map, $(b_{x1}, b_{y1})$ and $(b_{x2}, b_{y2})$ are the top-left corner and bottom-left corner of the predicted box.

## 4 EXPERIMENTS

### 4.1 IMPLEMENTATION DETAILS

Our approach is implemented in Python with Pytorch on 1 RTX2080Ti. The backbone is modified ResNet-50 as in (He et al. (2016)), and the weights are pre-trained on ImageNet (Russakovsky et al. (2015)). During the training phase, the model is optimized by the stochastic gradient descent (SGD), at the meantime, the total number of epochs is 20 and the batch size is set as 28. For the first 10 epochs, we frozen the parameters of the backbone and only train the heads structures, for the last 10 epochs, we unfrozen the last 3 blocks of backbone to be trained together. Besides, we warm up the training during the first 5 epoch with a warmup learning rate of 0.001 to 0.005, and in the last 15 epochs, the learning rate exponentially decayed from 0.005 to 0.00005.

### 4.2 RESULTS ON THREE BENCHMARKS

To affirm the effect of our method, we evaluate our tracker performance with the recent trackers MAML (Wang et al. (2020a)), PriDiMP (Danelljan et al. (2020)), SiamBAN (Chen et al. (2020)), SiamCAR (Guo et al. (2020)), ROAM (Yang et al. (2020)), SiamFC++ (Xu et al. (2020)), STN (Tripathi et al. (2019)), ARTCS (Kristan et al. (2019)), SiamRPN++ (Li et al. (2019)),SATIN (Gao et al. (2020)), ATOM (Danelljan et al. (2019)), DIMP-18 (Bhat et al. (2019)), DaSiamRPN (Zhu et al. (2018a)), SPM (Wang et al. (2019))ECO (Danelljan et al. (2017)), CFNet (Valmadre et al. (2017)), SiamRPN (Li et al. (2018)), DeepSRDCF (Danelljan et al. (2015a)), SRDCF (Danelljan et al. (2015b)) on four benchmarks UAV123 (Mueller et al. (2016)), VOT2018 (Kristan et al. (2018)), VOT2019 (Kristan et al. (2019)) and OTB100 (Wu et al. (2013))(details in Appendix A.1).

#### 4.2.1 RESULTS ON UAV123

UAV123 contains 123 challenging video sequences, which can be divided into 11 categories according to their attributes. The performance of the tracker is evaluated by two evaluation metrics, the precision scores and the AUC scores. The AUC scores reflect the overlap between the predict bounding box and ground-truth box, while the precision scores are relative with the distance between the center of the predict bounding box and ground-truth box. In Figure 3, our method achieves the best performance in precision scores, i.e., 0.850, and the second best AUC scores 0.678. As for the 11 categories of the challenge, SiamCAN ranks 1st or 2nd in Scale Variation, Similar Object, Fast Motion and Low Resolution, in Appendix A.2. The results demonstrate that our tracker can handle the similar object and scale change challenge, due to the learning of cross-attention subnetwork and anchor-free mechanism.

#### 4.2.2 RESULTS ON VOT2018

VOT2018 consists of 60 challenging videos. The evaluation metrics to rank the performance of the tracker based on EAO (Expected Average Overlap) which depends on the accuracy and the robustness. As shown in Table 1, our tracker outperforms SiamRPN++ by 2.8 points by introducing

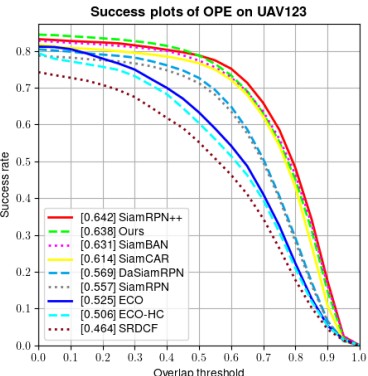 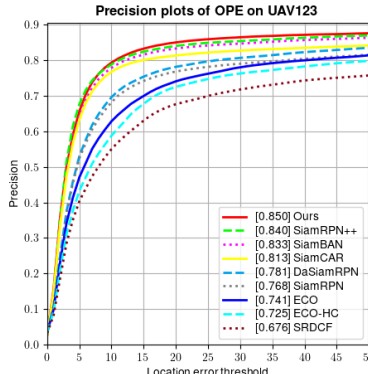

Figure 3: Success and precision plots on UAV123.

Table 1: Comparison with state-of-the-art trackers on VOT2018. Red, blue and green fonts indicate the top-3 trackers.

|  | SATIN | MAML | ATOM | SiamFC++ | SiamRPN++ | PrDiMP | SiamBAN | Ours |
|---|---|---|---|---|---|---|---|---|
| EAO↑ | 0.294 | 0.392 | 0.401 | 0.400 | 0.417 | 0.442 | 0.452 | 0.445 |
| A↑ | - | 0.635 | 0.590 | 0.556 | 0.604 | 0.618 | 0.597 | 0.591 |
| R↓ | - | 0.220 | 0.204 | 0.183 | 0.234 | 0.165 | 0.178 | 0.173 |

the anchor-free regression network. Compared with SiamFC++, we achieve EAO improvements of 4.5 points. Although our method fails to achieve the best EAO, the robustness scores of listed Siamese-based trackers are higher than our method, which affect the EAO scores, in other words, SiamCAN is a robust Siamese-based tracker.

### 4.2.3 RESULTS ON VOT2019

VOT2019 video sequences are 20% different from VOT2018, adding more fast motion and similar object videos. Table 2 reflects the evaluation results on VOT2019 compared with the recent trackers. We can see that the recent proposed MAML gets the highest accuracy scores, while our SiamCAN surpassing MAML by 2.5 points in terms of EAO. Besides, our robustness scores also rank 2nd.

### 4.3 ANALYSIS OF THE PROPOSED METHOD

**Discussion on effective sample selection.** Anchor-free method has the weakness that network may produce low-quality predicted bounding boxes far away from the center of target, even though the predicted box is accurate. To address this issue, SiamFC++ and SiamCAR introduce centerness branch to get high quality sample, forcing the network focus on the target center. While, SiamBAN uses ellipse figure region to design label which has the same effect. Accordingly, we do several experiments to find which method performs better. As shown in Table 3, baseline tracker consists of cross-attention module and anchor-free network. Ellipse label does better than Circle label (② vs ①),while the centerness branch with ellipse label even have worse effects (② vs ③). Based on

Table 2: Comparison with state-of-the-art trackers on VOT2019. Red, blue and green fonts indicate the top-3 trackers.

|  | SPM | ARTCS | SiamRPN++ | MAML | ATOM | STN | SiamBAN | Ours |
|---|---|---|---|---|---|---|---|---|
| EAO↑ | 0.275 | 0.287 | 0.292 | 0.295 | 0.301 | 0.314 | 0.327 | 0.331 |
| A↑ | 0.577 | 0.602 | 0.580 | 0.637 | 0.603 | 0.589 | 0.602 | 0.595 |
| R↓ | 0.507 | 0.482 | 0.446 | 0.421 | 0.411 | 0.349 | 0.396 | 0.376 |

Table 3: Results of different sample select methods on VOT2018.

| #Num | tracker | method | EAO |
|------|---------|--------|-----|
| ① | baseline | Circle label | 0.428 |
| ② | baseline | Ellipse label | 0.440 |
| ③ | baseline | Ellipse label+centerness | 0.436 |
| ④ | baseline | Ellipse&circle label | 0.445 |

Table 4: Results of each Components of our tracker on VOT2018.

| #Tracker | Components | Regression loss | EAO |
|----------|------------|-----------------|-----|
| ① | baseline | GIoU Loss | 0.351 |
| ② | baseline | DIoU Loss | 0.378 |
| ③ | + cross-attention | GIoU Loss | 0.411 |
| ④ | + cross-attention | DIoU Loss | 0.421 |
| ⑤ | + spatial-attention | DIoU Loss | 0.445 |

the performance of ②, we visualize the tracking of ② and compare with ④, details in Figure 4. At the training stage, ② gives the positive label to the points fall within $E_2$ region, while ④ gives the positive label to the points fall within the $C_2$ region, the more central position. In this aspect, the comparision in Table 3 (② vs ④) can be explained.

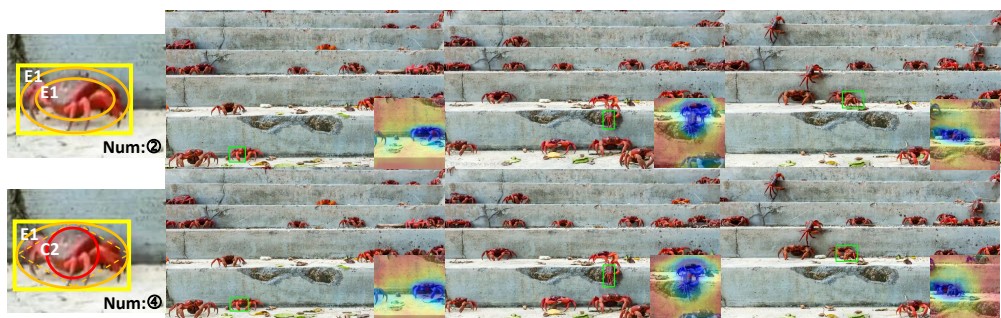

Figure 4: Comparision of different label assignments. Both ② and ④ predict accurate boundingbox, but the latter focuses more on the target center.

**Discussion on components of tracker.** To verify the role of each components of our tracker, we use the results of components on VOT2018 to analyze the details. As shown in Table 4, the baseline model can get an EAO of 0.351, which consists of a regular backbone, i.e., ResNet50, a classification and an anchor-free regression network. Exchanging DIoU Loss for GIoU Loss during training can get higher scores, due to the more accurate predicted bounding box (② vs ①). Adding cross-attention module can obtain a large improvement, i.e., 4.3 points on EAO (④ vs ②). This demonstrates the information interaction between the template branch and the search branch is of great significance. Finally, the tracker utilizes the self-spatial attention can gain another improvement of 2.4 points on EAO (④ vs ⑤).

**Feature visualization.** We visualize the features extracted from tracker ①, tracker ③ and tracker ⑤ in Figure 5. On the left side of Figure 5, tracker ① vs tracker ③ shows the tracker not equipped with cross-attention module is easier to lose target and focus on the wrong object when appear similar object and the visualized feature demonstrates that cross-attention module can enable tracker to tell target from similar distactors. On the right side of Figure 5, tracker ⑤ shows the power of cross-attention module combined with proper training strategy.

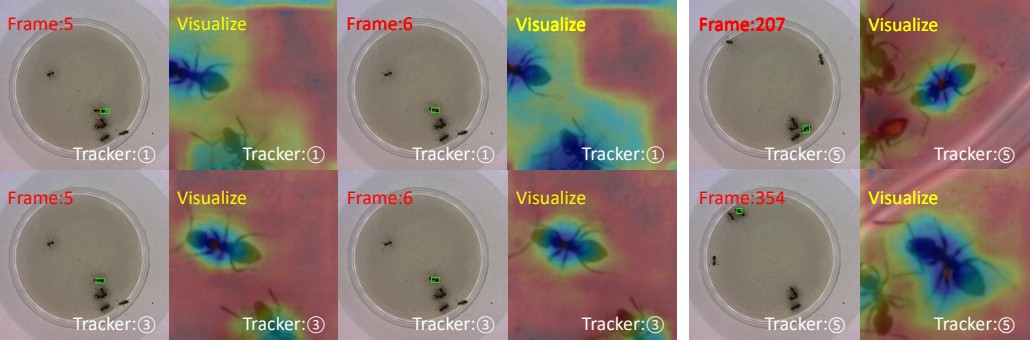

Figure 5: Visualization of confidence map. The left: the performance of tracker ① and tracker ③. The right: the performance of tracker ⑤.

## 5  CONCLUSION

In this paper, we propose a simple Siamese-based tracker called SiamCAN, which combined with cross-attention and anchor-free mechanism. The cross-attention module utilizes the target template channel attention to guide the feature learning of search frame, bridging the information flow between each branch. The anchor-free regression discards the fussy design of the anchors, and adjusts the scale ratio of the bounding box automatically. In order to use them to their fullest potential, choosing appropriate label assignment strategy and suitable loss funtion to boost the tracking performance with limited laboratory equipments. Extensive experiments are conducted on four benchmarks, which demonstrate our trackers with light structure yet achieves the state-of-the-art performance, especially in scale variation, background cluster, deformations and similar distractors challenges.

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

## A   APPENDIX

### A.1   EXPERIMENT RESULTS ON OTB100

OTB100 contains 100 challenging video sequences, and the evaluation metrics are same with UAV123. In Figure 5, our method achieves the best performance in precision scores, i.e., 0.913, and the second best success scores 0.684. As for the 9 categories of the challenge, SiamCAN ranks 1st or 2nd in Deformation, Background Clutters, Scale Varation and Out-of-Plane Rotation. The results demonstrate that our trackers can handle the deformation, background clutters, scale variation and out-of-plane rotation, in Figure 6.

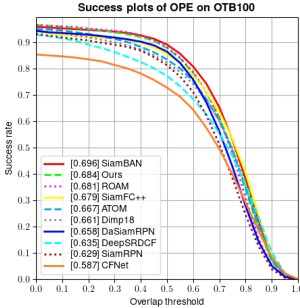 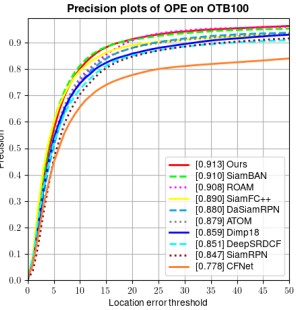

Figure 6: Success and precision plots on OTB100.

### A.2   MORE EXPERIMENT RESULTS ON UAV123

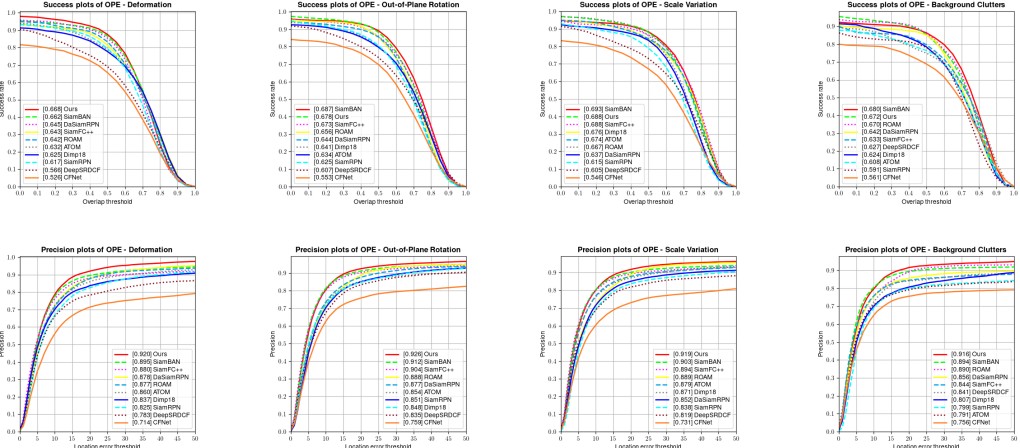

Figure 7: Comparisons on OTB-100 with challenges: Deformation, Background Clusters, Scale Variation and Out-of-Plane Rotation. Our SiamCAN performance ranks in top two.

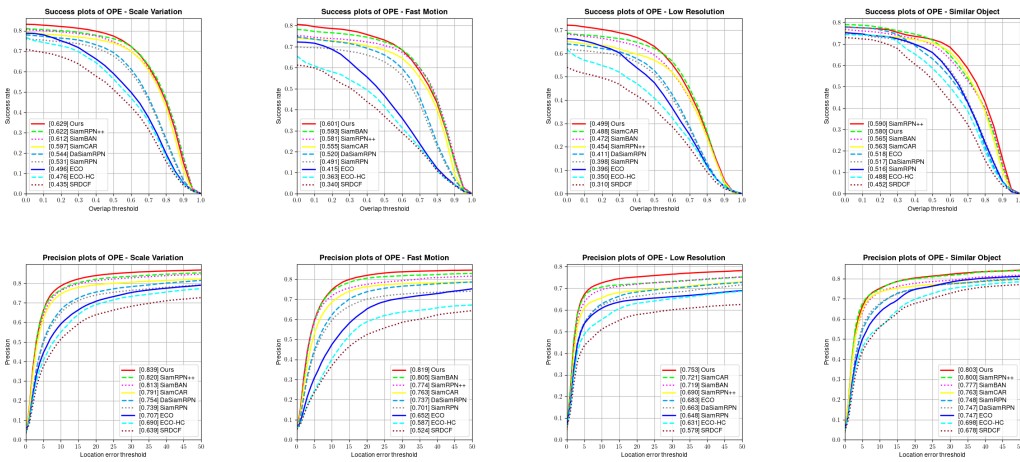

Figure 8: Comparisons on UAV123 with challenges: Scale Variation, Similar Object, Fast Motion and Low Resolution. Our SiamCAN performance ranks in top two.

