# OpenReview forum: "SiamCAN:Simple yet Effective Method to enhance Siamese Short-Term Tracking"
_ICLR.cc/2021/Conference — Reject_

### Official Review · AnonReviewer2 · 2020-10-25
**The proposed method is effective and technically sound, but lacks novelty and originality**

**Rating:** 4
**Confidence:** 5

**Review:**

#### 1.summary
In this paper, the authors introduce an cross-channel attrention mechanism and the anchor-free box regression branch with Diou-loss to deal with the clutters during the tracking procedure.
#### 2.strengths
The experimental results show that this method has excellent performance on several public benchmark datasets.
#### 3.weaknesses
- The novelty of the paper is deficient , the proposed method such as cross-attention mechanism [1], anchor-free regression [2, 3]  have been previously exploited in existing models.
- The proposed method is obviously based on the SiamBAN[2], however, there exist much reduplicate content which has already mentioned in [2].
- In experiments part, there are missing some important algorithms to compare, such as SiamBAN which is the baseline method for this paper.
- the analysis of the proposed method is deficient.

  *[1] Y. Yu, Y. Xiong, W. Huang, M. R. Scott, Deformable siamese attention networks for visual object tracking, in: Proceedings of the IEEE/CVF Conference on Computer Vision and Pattern Recognition, 2020, pp. 6728– 6737.*

  *[2] Zedu Chen, Bineng Zhong, Guorong Li, Zhang Shengping, and Ji Rongrong. Siamese box adaptive network for visual tracking. In IEEE Conference on Computer Vision and Pattern Recognition, pages 6668–6677,  2020.*

  *[3] Yinda Xu, Zeyu Wang, Zuoxin Li, Ye Yuan, and Gang Yu. SiamFC++: Towards robust and accurate visual tracking with target estimation guidelines. In AAAI, pages 12549-12556, 2020.*


#### 4. Correctness
The claims, method, and empirical methodology are correct.
#### 5. clarity
The paper writing is clear and easy to follow. But there exist some grammar faults. Moreover ,in sec 3.3, there are two same subtitles and similar content, which may be confusing. Besides, in figure 4, the arrangement of the pictures does not seem to match the description text.
#### 6. Relation to prior work
It is an incremental work based on the prior work.
#### 7. Reproducibility
Yes.

---

> ### Author Response · Authors · 2020-11-25
> **Response to AnonReviewer2**
>
> Thank you for your time and advises and references provided, about the weakness your pointed out:
>
> 1. "The novelty of the paper is deficient"
>
> Re: We acknowledge that attention mechanism and anchor-free regression have been previously exploited in existing models, however, compared to [2][3], our trackers with a light cross-attention module  outperforms them on several public datasets. Compared with [1], we design appropriate training strategy and use lighter structure (without mask learning, region refinement and deformable convolution, these are involved in Siamatten) to achieve the nearest performance on UAV123(ours AUC 0.638 Pr0.850;Siamatten AUC 0.650 Pr0.845), VOT2018(ours EAO 0.445 R:0.173;Siamatten EAO 0.470 R:0.16). I think how to use the simple method to achieve powerful performance is a kind of novelty. Beside, we add more analysis, experiments and comparision in the new paper, looking forward to your reading :).
>
> 2."The proposed method is obviously based on the SiamBAN"
>
> Re:In the new version, we add experiment to distinguish each difference. To be honest, our method is not similar with SiamBAN in details, hoping your reading again :).
>
> 3."In experiments part, there are missing some important algorithms to compare"
>
> Re: In the previous version, we compare the SiamBAN on UAV123 and OTB100 (in appendix). Based on your advises, we continue compare with it on VOT2018 and VOT2019. Moreover, we discuss the different between ours and SiamBAN in section 2 and 4 :).
>
> 4."The analysis of the proposed method is deficient."
>
> Re: Thanks for the references provided, we rewrite the section 2 and adding more ablation experiments in section 4.
>
> To sum up, thanks for your advises and we have uploaded the new paper hoping your reading and your valuable approval :).

---

### Official Review · AnonReviewer4 · 2020-10-26
**Reasonable method but contributions not enough**

**Rating:** 5
**Confidence:** 4

**Review:**

## Summary

This paper proposes a siamese network for visual tracking, namely SiamCAN, which utilizes cross channel attention, spatial attention, and anchor-free regression head. The method achives state-of-the-art performance on four visual tracking benchmarks.

## Strengths

 +. The design of the network is reasonable, including using template's channel information to help the search branch to learn more specific feature, using spatial attention to aggregate location information from the depth-wise correlation, and utilizing anchor-free regression head to locate the object.

+. The final results are good, and ablation study shows the effectiveness of these modules.

## Weaknesses

-. The technical contribution is weak. The main components, including channel attention, spacial attention and the anchor free network, are not new. This proposed method is quite similar to SiamBAN, especially the multi-head fusion and the anchor free network.  Furthermore,  the cross channel attention is more like a cross-correlation between the search branch feature and the pooling template vector. In this case, are the 1D conv layer and the sigmoid function necessary?

-. The description of the method is not clear. In Figure 2, there is only one correlation map but in Sec. 3.3 the author sad it has N correlation maps from different layers of the backbone. Furthermore, the index "i" of these N layers is conflict with the position index "point(i, j)", which would lead to misunderstanding. Also in Figure 2, the classification map's size is 25x25x1, but it is 25x25x2 in Sec. 3.3. The description of spatial attention is not clear. We can find Max/AvgPool in the figure but find no description. So how is the spatial attention used? The only word I can find is in related works "CBAM (Woo et al. (2018)) utilizes both max-pooling and average-pooling to generate the merged attention, includes channel and spatial attention". Is that the same? In Sec 3.2, "Inspired by (Wang et al. (2020b)), we add channel attention and spatial attention into our network. ". However, Wang et al. (2020b) did not use spatial attention module.

## Overall Rating

Due to the weakness described above, I rate a "Marginally below acceptance threshold" score for the paper.

---

> ### Author Response · Authors · 2020-11-25
> **Response to AnonReviewer4**
>
> Thank you for your time and important question about our work. We address several points of your evaluation:
>
> 1."The technical contribution is weak."
>
> Re: I have to admit that the attention mechanism and anchor free appoarch are not new, but how to use them to gain powerful performance is what we should focus on , from our perspective. It has been a long time that  tracking reasearch have to depends on sufficient Laboratory equipment such as numerous GPUS to gain better performance. How to use limited sources to have ideal results is of great importance for tracking community. Our tracker adopts simple yet effective method to gain nearest SOTA results.  In this aspect, I think our paper has quiet technical contribution. I'm so sorry that our first paper is not satisified due to the limited time. We have added more analysis, experiments and our contribution in the new submit, looking forward to your reading  :).
>
> 2."The cross channel attention is more like a cross-correlation between the search branch feature and the pooling template vector. In this case, are the 1D conv layer and the sigmoid function necessary?"
>
> Re: We think it has difference bewteen the cross-correlation and cross channel attention, the former tends to calculate the similarity between two branches which is a recify stage, while the latter tends to propagate the information from template breach to search branch which is a learning stage. The 1D conv layer and the sigmoid function your mentioned is a part of ECA network which is involved in cross attention module :).
>
> 3."The description of the method is not clear"
>
> Re: I'm so sorry again, due to our careless, the figure and caption are confusing. Based on the problem you pointed out, we upload the new figure2, add the discription about spatial attention and delete the irrelavent reference in section 3.2 :)
>
>
> To sum up, thanks for your advises and we have uploaded the new paper hoping your reading and your  valuable approval :).

---

### Official Review · AnonReviewer1 · 2020-10-27
**This paper presents a Siamese-based single object tracking method using attention mechanism both in channel-wise and spatial-wise for learning deep correlation between exemplar and candidate images.**

**Rating:** 3
**Confidence:** 5

**Review:**

This paper presents a Siamese-based single object tracking method using attention mechanism both in channel-wise and spatial-wise for learning deep correlation between exemplar and candidate images. Extensive experiments on UAV123, VOT2018 and VOT2019 demonstrate the effectiveness of the proposed method, and the 35 FPS running speed indicates its competitive efficiency.

Strengths:
1. The idea of multiplying the weights of aggregated template feature to the search feature is interesting and rather intuitive, the experimental results on several datasets confirmed the effectiveness.
2. This paper is well-organized and easy to follow.
3. Comprehensive experiments including the appendix are carried out, and convincing.

Weaknesses:
1. Lacking important literature reviews on highly relevant works.
The proposed SOT method belongs to 1) anchor-free and 2) attention-based pipelines. There are several latest and popular works the author should mention in Related Works:
SiamFC++: Towards Robust and Accurate Visual Tracking with Target Estimation Guidelines   https://arxiv.org/pdf/1911.06188.pdf
SiamCAR: Siamese Fully Convolutional Classification and Regression for Visual Tracking    https://arxiv.org/abs/1911.07241v2
SATIN: Siamese Attentional Keypoint Network for High Performance Visual Tracking
https://arxiv.org/pdf/1904.10128.pdf
All three works above are anchor-free methods, and the third paper SATIN adopts attentional mechanism for SOT.
The author compared the proposed method with SiamFC++ and SiamCAR in experimental results, but the review of these methods and explanation of difference in methodology are missing in this paper.
As a matter of fact, the most relevant work to this paper is SATIN which also adopted channel and spatial attention in Siamese architecture. The only difference is multiplying the attentive channel weights of exemplar to candidate, which is called cross-attention in this paper.
The author should give a more thorough review and explain the difference between these works.

2.  The contribution of this work is rather limit.
The paper describes three contributions in section 1, however,  the second point (anchor-free and DIoU) is hardly the contribution of this work  but reimplementation of previous works, and the third point is only a report of experimental results. In fact, the only contribution of this work is introducing cross-channel attention to explicitly learn the correlation of template and search images, which is a small trick in network design. Compared to SATIN, which also adopts channel attention to template feature and spatial attention later,  this paper does not bring a substantial contribution to visual tracking community.

3. The comparison with SOTA methods are not comprehensive.
The experimental results on UAV123 do not include SOTA method SiamFC++, similarly, results on VOT2018 do not include SiamCAR, and results on VOT2019 do not include SiamFC++ and SiamCAR. Moreover, the author should also compare the results with SATIN on both OTB and VOT datasets.

4. The presentations of Figure 1 and Figure 4 are terrible.
The resolution of Figure 1 is rather low, and the author did not even explain which one corresponds to the challenge situation of three aspects in figure caption. It is hard to tell which one is better in case 1 (first row) and case 2 (second row) due to the low resolution.
Figure 4 is rather confusing. The author did not explain the visualized confidence map belongs to which layer, and this attentive map is whether from template feature or search feature. I assume the second row on left side is from the subsequent frames, then what’s the meaning of both rows on right side, and why there is a blank gap between left and right?
The author should pay more attention to improve the figures in this paper, and the captions should be more thorough without ambiguity.

---

> ### Author Response · Authors · 2020-11-25
> **Response to AnonReviewer1**
>
> Thank you for your valuable feedback!Please see our point-by-point response below:
>
> 1. "Lacking important literature reviews on highly relevant works. "
>
> Re: We acknowledge that our paper lacks highly relevant anchor-free trackers, such as SiamFC++, SiamBAN, SiamCAR and attention-based tracker, SATIN. We add a more thorough review in section 2 and have submitted the new paper, looking forward for your reading. About the cross-attention module in SATIN, we think it just has the same name with ours. The 'cross' in SATIN means attention cross shallow and deep layers, while ours 'cross' tends to represent the information flow between the search branch and template branch which is totally different. Don't worry, the above analysis is rewritten in the paper :).
>
> 2."The contribution of this work is rather limit."
>
> Re: Due to the paper first submit time, we haven't  done a good writing to emphasis our contribution. In the new paper, we add more experiments and analysis about our works. Our goal is to use the simple strategy and light structure to achieve nearest to the SOTA performance. We hope our work can be adopted by other reseachers as strong baseline like SiamFC, SiamRPN. From this prospective, we think our work can do a potential contribution to tracking community :).
>
> 3."The comparison with SOTA methods are not comprehensive."
>
> Re: As for the reason of  experimental results on UAV123 and on VOT2019 do not include SiamFC++ , the official website  https://github.com/MegviiDetection/video_analyst/blob/master/docs/TUTORIALS/SOT_SETUP.md  and paper don't provide their raw results, so I‘m regret comparing tracking results without this powerful tracker too :(. The same reason for SiamCAR  https://github.com/ohhhyeahhh/SiamCAR . About SATIN, the code haven't released so the only approach to get the tracking result is the data in paper. All in all, I add the more relative tracking results in the new paper as much as I could find :) .
>
> 4. "The presentations of Figure 1 and Figure 4 are terrible."
>
> Re: I'm so sorry that our pics and caption are so confusing, we already upload the high resolution figure with clear annotation  and rewrite the caption.
>
> To sum up, thanks for your patient reading. We have rewrite the paper according to your advise, hoping it can get your valuable approval :).

---

### Official Review · AnonReviewer3 · 2020-10-29
**A siamese short-term tracker with a cross-guided network is introduced.**

**Rating:** 3
**Confidence:** 4

**Review:**

The architecture of the tracker is standard siamese. The novelty is at a technical level, modules of the "cross-guided" type have been proposed. It does bring an improvement, but not to the state-of-the-art level.  There is no significant insight, training, updating novelty or theoretical. Recent short-term trackers output segmentation, the proposed tracker outputs a bounding box.

The performance of the tracker is evaluated on UAV, VOT 2018 and VOT 2019. UAV is saturated. The performance VOT 2019 is worse than state-of-the-art, which is not reported -- the trackers selected for comparison do not include the best performing ones. It is not clear why  VOT 2020 was not included.

Overall, this is "yet another siamese tracker", which is not sufficient for ICLR acceptance.

---

> ### Author Response · Authors · 2020-11-25
> **Response to AnonReviewer3**
>
> Thank you for your valuable feedback! Concerning the points raised in the your review:
>
> 1."There is no significant insight, training, updating novelty or theoretical. "
>
> Re: We're so sorry that the first submitted version didn't clearly explain our novelty, due to the limited time. We have submitted the new version which involves more experiments , analysis and comparision with other anchor-free trackers. Briefly speaking, the most novelty of our paper is how to use the existed method and limited sources (such as one GPU) appropriately to achieves the nearest to the SOTA performance.  Honestly, it has been a long time that tracking research have to depends on sufficient Laboratory equipment, the more GPUS the better performance. So I think our methods have a potential contribution to tracking community to some extent. More details looking forward your reading :).
>
>  2. " The performance VOT 2019 is worse than state-of-the-art, which is not reported"
>
> Re: As we known, the best performance in VOT2019 is Ocean (EAO:0.350 R:0.316 ), while it involves a update branch. However, Ocean without update branch(EAO:0.327 R:0.376 ), ours (EAO:0.330 R:0.376 ) ,  the gap between the result of two trackers is not very big. Moreover, Oceans involves  feature alignment module and feature combination  module,  which is more complex than our module and have larger layer parameters.
>
> 3. "The trackers selected for comparison do not include the best performing ones"
>
> Re: We acknowledge that some trackers we don‘t compared in our paper, such as Ocean, Siamatten. The reason is that it is not easy for us to compare with them, for example, both of them use the deformable convolution but they don't do ablation study about that part so I cannot analysis at the same level. However, our tracker also achieves not bad results, for example,  on UAV123(ours AUC 0.638 Pr0.850;Siamatten AUC 0.650 Pr0.845), VOT2018(ours EAO 0.445 R:0.173;Siamatten EAO 0.470 R:0.16). And our method not always get the best scores, details in new paper section 4.
>
> 4."It is not clear why VOT 2020 was not included."
>
> Re: We're so sorry that due to the limited time and sources, we don't have sufficient time to test and analysis on VOT2020. We will finish this work later :).
>
> 5."This is "yet another siamese tracker""
>
> Re: We acknowledge that our baseline is siamese tracker, but our goal is not to reproduce another Siamese tracker. The goal of our paper is to discuss if we use appropriate method  and training strategy  can we get satisfactory result. The answer is yes. The simpler network structure, the better performance. From this prospective, I don't think our work is yet another Siamese  tracker :).
>
> To sum up, thanks for your patient reading. We have rewrite the paper according to your advise, hoping it can get your valuable approval :).

---

### Decision · Program_Chairs · 2021-01-07
**Final Decision**

**Decision:**

Reject

**Comment:**

All three reviewers initially recommended reject.  The main concerns were:
1) weak technical contribution and insight [R1, R2, R3, R4];
2) incremental novelty (another variation of SiamFC) [R1, R2, R3];
3) unconvincing experiment results against missing SOTA [R1, R2, R3];

The author's response did not assuage these concerns.